# Evolving Indications for Liver Transplantation for Hepatocellular Carcinoma Following the Milan Criteria

**DOI:** 10.3390/cancers17030507

**Published:** 2025-02-03

**Authors:** Takashi Kokudo, Norihiro Kokudo

**Affiliations:** National Center for Global Health and Medicine, Tokyo 162-8655, Japan; nkokudo@hosp.ncgm.go.jp

**Keywords:** hepatocellular carcinoma, liver transplantation, Milan criteria

## Abstract

Since their introduction in the 1990s, the Milan criteria have been the gold standard of indication for liver transplantation (LT) in patients with hepatocellular carcinoma (HCC). Nevertheless, several institutions have reported wider indication criteria for LT with comparable survival outcomes. This paper summarizes the recent indications for LT for HCC through a literature review.

## 1. Introduction

Liver transplantation (LT) is the theoretically ideal treatment modality for hepatocellular carcinoma (HCC) to avoid recurrence [1,2]. Nevertheless, recurrence occurs even after LT, and the prognosis is dismal in patients with post-transplant recurrence [3]. Hence, the selection criteria of the recipient are essential to take complete advantage of the therapeutic benefits of LT.

The Milan criteria are the most well-known size and number criteria, and first reported in 1996 [4]. The criteria were a tumor measuring ≤ 5 cm in diameter in patients with a single HCC, and no more than three tumor nodules each measuring ≤ 3 cm in diameter in patients with multiple tumors. After the turn of the century, multiple new expanded criteria were reported with acceptable survival outcomes (Table 1) [5]. In this review, we provide an overview of the reported criteria for LT in patients with HCC.

The Milan criteria depend on morphological parameters, and provide an opportunity for LT to only 30% of the patients with HCC [6]. Nevertheless, various observations have demonstrated that there are patients with tumors that are beyond the Milan criteria who have favorable outcomes after LT [7]. The idea of expanding beyond the Milan criteria spread rapidly among the transplant community [5].

Two types of LT are performed worldwide. The first one is deceased donor liver transplantation (DDLT), which is primarily performed in Western countries. Conversely, living donor liver transplantation (LDLT) is a major modality because of the shortage of deceased donors in the East [8]. Because LDLT has donor risks, the transplantation criteria should be stricter than those for DDLT [9].

**Table 1 cancers-17-00507-t001:** Summary of the eligibility criteria for liver transplantation in hepatocellular carcinoma.

Eligibity Criteria	Year	Criteria
Based on tumor number and size only		
Milan [4]	1996	Single tumor ≤ 5 cm ≤3 tumors, all ≤ 3 cm
UCSF [10]	2001	Single tumor ≤ 6.5 cm ≤3 tumors, all ≤ 4.5 cm Total tumor diameter ≤ 8 cm
5-5 rule [11]	2007	Tumor number ≤ 5 Maximum diameter ≤ 5 cm
Up-to-seven [12]	2009	Sum of the size of the largest tumor [in cm] and the number of tumors ≤ 7
Based on number and size, plus tumor markers		
Kyoto [13]	2007	Tumor number ≤ 10 Maximum diameter ≤ 5 cm Serum DCP level ≤ 400 mAU/mL
French AFP model [14]	2012	Score (model including tumor size, tumor number, and AFP level) ≤ 2
TTV/AFP [15]	2015	TTV ≤ 115 cm^3^ and AFP level ≤ 400 ng/mL
Metroticket 2.0 [16]	2018	Sum of the size of the largest tumor [in cm] and the number of tumors ≤ 7 + AFP level < 200 ng/mL Sum of the size of the largest tumor [in cm] and the number of tumors ≤ 5 + AFP level 200–400 ng/mL Sum of the size of the largest tumor [in cm] and the number of tumors ≤ 4 + AFP level 400–1000 ng/mL
5-5-500 rule [17]	2019	Tumor size ≤ 5 cm in diameter Tumor number ≤ 5 AFP level ≤ 500 ng/mL
Based on tumor differentiation		
Toronto [18]	2011	Any size or number without extrahepatic disease vascular invasion poorly differentiated Tumor
Extended Toronto [19]	2016	Any size or number without systemic cancer-related symptoms extrahepatic disease vascular invasion poorly differentiated tumor

Abbreviations: AFP, alpha-fetoprotein; UCSF, University of California San Francisco; BCLC, Barcelona Clinic Liver Cancer; DCP, des-gamma-carboxy prothrombin; TTV, total tumor volume.

## 2. Expanding the Milan Criteria by Tumor Size

After the report of the Milan criteria, Yao et al. reported the University of California San Francisco (UCSF) criteria in 2001, where in the tumor size was expanded as follows: solitary tumor ≤ 6.5 cm, or three or fewer nodules with the largest lesion measuring ≤ 4.5 cm, and the total tumor diameter measuring ≤ 8 cm [10]. These patients had survival rates of 90% and 75.2% at 1 and 5 years, respectively, after LT, compared to a 50% 1-year survival rate for patients with tumors exceeding these limits (*p* = 0.0005). These criteria expanded the Milan criteria by tumor size, but still preserved excellent survival after LT.

## 3. Expanding the Milan Criteria by Tumor Number

The limit of tumor number is three in the Milan criteria, and several reports have extended this limit. The relatively well-known criteria for tumor number are the so-called “Up-To-Seven” criteria [12], in which the inclusion criterion was “HCC with seven as the sum of size of the largest tumor (in cm) and the number of tumors”. This criterion was also reported by Mazzaferro V, who initially proposed the Milan criteria. Besides this report, the Tokyo group expanded the tumor number to five [11]. The criteria reported by the Tokyo group are based on LDLT; however, they reported that recurrence-free survival rates at 3 years for patients fulfilling the criteria and for those exceeding the criteria were 94% and 50%, respectively, and concluded that the indication for HCC might be expanded from the Milan criteria with equivalent outcomes.

## 4. Expanding the Milan Criteria by Biomarkers

In addition to tumor number and size, several biomarkers have been used to evaluate tumor aggressiveness. Alpha-fetoprotein (AFP) is the most widely used biomarker for HCC, and it was added to several criteria. Mazzaferro V also reported a Metroticket 2.0 model that included not only tumor size and number, but also serum AFP level [16]. The model identified patients who survived for 5 years after liver transplantation with an accuracy of 0.721 (95% confidence interval, 0.648−0.793%), which was higher than the predictive abilities of the Milan, UCSF, and Up-To-Seven7 criteria (*p* < 0.001). The French AFP model also included the serum AFP level [14]. Toso C et al. reported criteria that included total tumor volume and AFP level [15]. The Japanese nationwide survey also reported a 5-5-500 rule, including the serum AFP level [17]. The Kyoto group developed criteria using protein induced by vitamin K absence or antagonist-II (PIVKA-II), also known as des-gamma-carboxy prothrombin (DCP) levels [13].

## 5. Expanding the Milan Criteria by Tumor Differentiation

DuBay et al. reported the Toronto criteria in 2011, according to which a pretransplant tumor biopsy is performed, and irrespective of the tumor size and number, LT could be performed for patients with HCC not poorly differentiated on biopsy [18]. These innovative criteria expand the indication of liver transplantation (LT) regardless of the tumor number and size, although tumor biopsy before LT should be performed. The same group reported extended Toronto criteria in 2016, which included the absence of cancer-related symptoms [19]. The criteria required tumor biopsy to select patients for LT.

## 6. Impact of Microvascular Invasion in Patients with HCC

Several retrospective studies have reported microvascular invasion (MVI) as a significant risk factor for postoperative recurrence [20,21,22,23]. Therefore, predicting MVI preoperatively is important in both LT and liver resection because of its high risk of recurrence. Several studies have attempted to identify preoperative predictive factors for MVI using tumor markers or preoperative imaging findings [24,25,26]. The reported preoperative imaging findings include computed tomography (CT) [27], contrast-enhanced magnetic resonance imaging (MRI) [28,29,30,31], and ^18^F-fluorodeoxyglucose positron emission tomography (FDG-PET) [31,32].

Peng et al. [33] constructed a radionics nomogram using CT images to predict MVI in patients with HCC related to hepatitis B virus (HBV). They found that the radionics nomogram, as a noninvasive preoperative prediction method, demonstrated favorable predictive accuracy for the MVI status in patients with HBV-related HCC.

Three MRI features were reported to be independently associated with MVI, viz., arterial peritumoral enhancement, non-smooth tumor margin, and peritumoral hypointensity in the hepatobiliary phase [29]. Shirabe et al. proposed a new scoring system for predicting MVI, which consisted of tumor size, serum DCP levels, and the maximum standardized uptake value on FDG-PET [24].

Compared with CT/MRI, ultrasound (US) is radiation-free, easy to implement, and simple to use for liver examinations. Hu et al. [34] reported a US-based radionics score for the preoperative prediction of MVI. They found AFP level, tumor size, and radionics score as independent predictors of MVI. The radionics nomogram demonstrated better performance in detecting MVI than the clinical nomogram.

High glucose consumption by the tumor cell microenvironment, including that in HCC, can be reflected in FDG-PET. Although the efficiency of tumor detection is affected by liver cirrhosis and a high background signal, a recent study [35] reported that the FDG-PET/CT radionics signature was an independent biomarker for MVI estimation.

## 7. Bridging Therapy Before LT

Patients with HCC who are listed for LT are often treated while on the waiting list to prevent the progression of HCC or reduce the measurable disease burden of HCC [36,37]. This strategy is applied to patients who do not fulfill the criteria. Table 2 summarizes the techniques used for bridging or downstaging [38]. Despite the lack of data from randomized controlled trials, the recent European guidelines recommend neoadjuvant therapies to reduce the dropout risk due to tumor progression [39]. The recent guidelines from the American Association for the Study of Liver Diseases also suggest bridging therapy [40]. Transarterial chemoembolization (TACE) shows good outcome while on the waiting list with an excellent outcome after LT, with the 1-, 2-, and 5-year survival rates being 98%, 98%, and 93%, respectively [41].
cancers-17-00507-t002_Table 2Table 2Summary of bridging therapies before liver transplantation in hepatocellular carcinoma.Bridging TherapyYearFirst Author5-Year Survival After TransplantationTACE [41]2003Graziadei IW93%Radioembolization [42]2006Kulik LM27% *RFA [43]2005Lu DS76% *SBRT [44]2017Sapisochin G75%Resection [45]2003Belghiti J61%Immunotherapy [46]2024Tabrizian P85% ** 3-year survival. Abbreviations: TACE, transarterial chemoembolization; RFA, radiofrequency ablation; SBRT, stereotactic body radiotherapy.

Radioembolization with yttrium-90 microspheres has been shown to limit tumor progression and dropout from LT [42]. Kulik et al. reported that 23% of the patients were successfully downstaged and underwent LT after treatment. The 1-, 2-, and 3-year survival rates in these patients were 84%, 54%, and 27%, respectively.

Radiofrequency ablation (RFA) has been reported as a bridging therapy, with the 1- and 3-year post-transplantation survival rates being 85% and 76%, respectively [43]. Lu et al. concluded that percutaneous RFA is an effective bridging therapy to LT for patients with compensated liver function and safely accessible tumors.

There is limited experience with stereotactic body radiotherapy (SBRT) as a bridging therapy; however, one study compared SBRT with TACE or RFA and found similar results among the groups [44]. The 1-, 3-, and 5-year survival rates from the time of transplantation were 83%, 75%, and 75% in the SBRT group compared to 96%, 75%, and 69% in the TACE group and 95%, 81%, and 73% in the RFA group, respectively (*p* = 0.7).

Another strategy may be used for selected patients with small tumors and adequate liver function, which consists of initial surgical tumor resection followed by close surveillance and “salvage” transplantation if tumor recurrence or deterioration of liver function occurs. A retrospective study of patients with HCC undergoing LT compared with patients subjected to initial resection and then LT reported that the postoperative course and overall survival were not different [45]. The study concluded that, in selected patients, liver resection before transplantation does not increase the morbidity or impair long-term survival after LT. Therefore, liver resection before transplantation can be integrated into the treatment strategy for HCC.

Immune checkpoint inhibitor (ICI) is recommended as a first- and second-line treatment for unresectable or advanced HCC [40]. A recent study reported the safety and outcomes of neoadjuvant immunotherapy before LT, with a 3-year post-LT survival rate of 85% [46]. Graft rejection is the major concern with pretransplant ICI use. In their series, 16.7% of patients experienced rejection; however, no deaths occurred secondary to graft rejection. They concluded that patient with HCC who received ICI pre-LT showed favorable survival and safety outcomes, justifying continued use.

## 8. Discussion

This review has summarized the criteria for LT for patients with HCC. The initial Milan criteria were reported in 1996, which were subsequently expanded by several centers, with the outcomes being comparable to those of patients included within the Milan criteria. Besides tumor size and number, which are included in the Milan criteria, recent criteria included biomarkers such as AFP and tumor differentiation.

The relatively well-known criteria that included AFP is the Metroticket 2.0 model proposed by Mazzaferro et al. [16]. The efficiency of this model depends on the inclusion of AFP in the equation predicting post-transplantation prognosis in HCC. HCC results in a “double prognosis” depending on two major components, viz., tumor burden and liver function. This implies that it is difficult to determine cancer-specific outcomes in such patients, considering the impact of nontumoral conditions on patient performance status, eligibility to therapies, risk of de novo tumors, and ultimately survival. The liver function status should be considered to determine the indication of LT. In Italy, the model for end-stage liver disease score is used as the criteria for LT [47].

The presence of MVI in patients with HCC is a significant prognostic factor after LT [20]. Therefore, it is important to predict MVI before HCC treatment. If MVI presence is anticipated, the patient might not be a good candidate for LT, considering the high incidence of HCC recurrence after LT. From this viewpoint, the prediction of MVI is essential to determine therapeutic modalities. Because MVI has also a significant risk of recurrence after liver resection [48], there have been several attempts to predict MVI preoperatively [33,34,35]. However, since the preoperative prediction of MVI is difficult, including MVI in the indication criteria would have limited value.

Bridging therapy is an attractive strategy to perform locoregional treatments to bring patients whose tumor burden does not fulfill the criteria [49]. In a prospective, uncontrolled study, tumor downstaging was beneficial, with the post-transplantation outcomes being comparable to those of patients included within the criteria [50]. As shown in Table 2, TACE, radioembolization, RFA, SBRT, and surgical resection are the reported bridging therapies before LT [41,42,43,44,45,46]. Prospective trials investigating tumor downstaging as a tool to expand the conventional criteria for LT are required and would be a future direction of expanding the indication of HCC before LT.

Considering the shortage of deceased donor organ pool and the ethical difficulty with living donors, it is necessary to develop ideal criteria that can be universally used. Although the Milan criteria are the most commonly used criteria, there is increasing evidence that they can be expanded with comparable outcomes. Adding biomarkers may be a promising direction to appropriately transplant organs to the appropriate recipient. Future studies will be required for developing ideal criteria for LT to prevent post-transplant recurrence. Bridging therapy is another approach to bring patients whose tumor burden does not fulfill the criteria; however, several studies are limited by the small sample size and short follow-up duration [50,51,52,53,54]. Future prospective trials would be required to clarify the benefit of bridging therapy.

The current European Association for the Study of the Liver guidelines and the American Association for the Study of Liver Diseases (AASLD) guidelines recommend the Milan criteria for the indication of LT [55,56]. In the AASLD criteria, the UCSF criteria, total tumor volume cutoff of 115 cm, Up-To-Seven criteria, extended Toronto criteria, and Kyoto criteria are also proposed as the expanded criteria [56]. In the guidelines for the Asia–Pacific region, the Milan criteria are also proposed and the expanded criteria are considered limited due to the scarcity of the donor pool [57].

## 9. Conclusions

There are several criteria expanding Milan criteria, with the outcomes being comparable to those of patients included within the Milan criteria. Besides tumor size and number, which are included in the Milan criteria, recent criteria included biomarkers and tumor differentiation.

## Data Availability

There is no new data created for this paper.

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
