# Peer review of "Evolving Indications for Liver Transplantation for Hepatocellular Carcinoma Following the Milan Criteria"

_cancers, 2025, doi:10.3390/cancers17030507_

Round 1
Reviewer 1 Report
Comments and Suggestions for Authors
1- table 1 is not well designed and is not easy to be read, since the criteria data does not fit with each eligible criteria, it should be better organized.
2- in the area where it is explained who the Milan criteria is expanded by tumor numbers( it is not clear that criteria of up to seven), then in the biomarkers (it is not clear what levels are relevant, no data about outcomes using the criteria), then when is based on tumor characteristics, it is not clear based on tumor differentiations and how it should be managed in the pre op setting
3- in the subject of bridging therapy before transplant, authors mentioned the European data with neoadjuvant therapy, but it is not clear the recommendations, is that new adjuvant for all pts with HCC? or for certain particularly patients?
4- in the area of discussion .. it is not clear the concept of the "double prognosis" and the importance in the outcomes. i would explain in more e details of that concept.
also in the discussion area, I would explain in more details about MVI and ways to predict it and how it is proposed in the pre transplant setting based on current data.
Author Response
- table 1 is not well designed and is not easy to be read, since the criteria data does not fit with each eligible criteria, it should be better organized.
Response: We thank the Reviewer for their helpful comment. According to the second comment of Reviewer 2, we have categorized the criteria based on tumor number and size, tumor marker levels, and tumor characteristics.
- in the area where it is explained who the Milan criteria is expanded by tumor numbers( it is not clear that criteria of up to seven), then in the biomarkers (it is not clear what levels are relevant, no data about outcomes using the criteria), then when is based on tumor characteristics, it is not clear based on tumor differentiations and how it should be managed in the pre op setting
Response: We thank the Reviewer for their salient comment. The up-to-7 criteria include the criterion of more than 3 nodules; therefore, the number is considered to be expanded. We have added the data related to Metroticket 2.0 and its superiority to the revised manuscript as follows (page 3, lines 27–30): “The model identified patients who survived 5 years after liver transplantation with an accuracy of 0.721 (95% confidence interval, 0.648%−0.793%), which was higher than the predictive abilities of the Milan, UCSF, and up-to-7 criteria (P < .001)”. We have also clarified the novelty of the criteria based on tumor differentiation as follows (page 3, lines 39–41): “These innovative criteria expand the indication of liver transplantation (LT) regardless of the tumor number and size, although tumor biopsy before LT should be performed.”
- in the subject of bridging therapy before transplant, authors mentioned the European data with neoadjuvant therapy, but it is not clear the recommendations, is that new adjuvant for all pts with HCC? or for certain particularly patients?
Response: We thank the Reviewer for their careful observation. Bridging therapy should be utilized in patients who do not fulfill the criteria. We have revised page 4, line 26 as follows: “This strategy is applied to patients who do not fulfill the criteria.”
- in the area of discussion .. it is not clear the concept of the "double prognosis" and the importance in the outcomes. i would explain in more e details of that concept.
also in the discussion area, I would explain in more details about MVI and ways to predict it and how it is proposed in the pre transplant setting based on current data.
Response: We thank the Reviewer for their comment. We have added the following statement to state the importance of liver function status (page 5, lines 36–37): “The liver function status should be considered to determine the indication of LT.” We have also added a new reference (#46). Although we overviewed the predictors of MVI, there was no complete way to predict MVI. We have clarified this point by adding the following to the revised manuscript (page 5, lines 45–46): “However, since the preoperative prediction of MVI is difficult, including MVI in the indication criteria would have limited value.”
Reviewer 2 Report
Comments and Suggestions for Authors
This original manuscript arouses interest for readers and provides an important clue to identifying candidates with hepatocellular carcinoma for liver transplantation using the optimal criteria. However, several issues should be addressed or altered.
1) The "SIMPLE SUMMARY" and "ABSTRACT" sections are almost identical, which raises concerns about their distinct purposes. Please revisit the journal's guidelines and clarify the intended differences between these sections. A rewrite emphasizing these differences is strongly recommended.
2) Table 1 is visually cluttered and somewhat difficult to read. Specifically, the "Criteria" column lacks clear differentiation from other studies. You could group similar criteria together for better visual organization.
3) The order of the eligibility criteria in Table 1 is unclear. Are the criteria arranged chronologically, alphabetically, or with/without biomarkers? Please specify the rationale behind this arrangement in the table legend. If no specific order is followed, reorganizing the criteria based on clinical relevance or frequency of use could enhance readability and utility.
4) Listing the first author in Table 1 adds limited value. Instead, prioritize summarizing clinically relevant data, such as the number of patients analyzed, outcomes, or the breakdown of DDLT vs. LDLT. If space is limited, it is advisable to omit the first author entirely to focus on more critical information.
5) The manuscript does not adequately address the prevailing trends in the use of eligibility criteria. To strengthen the discussion, provide an overview of the current global practices. Discuss which criteria are predominantly used today, with specific attention to regional variations—e.g., in Europe, North America, and Asia. Incorporating a clear summary of these trends, supported by relevant data or literature, would add substantial value to the manuscript.
Author Response
This original manuscript arouses interest for readers and provides an important clue to identifying candidates with hepatocellular carcinoma for liver transplantation using the optimal criteria. However, several issues should be addressed or altered.
- The "SIMPLE SUMMARY" and "ABSTRACT" sections are almost identical, which raises concerns about their distinct purposes. Please revisit the journal's guidelines and clarify the intended differences between these sections. A rewrite emphasizing these differences is strongly recommended.
Response: We thank the Reviewer for their comment. We have modified the abstract by adding the following sentences: “Since their introduction in the 1990s, the Milan criteria have been the gold standard for LT indication in patients with hepatocellular carcinoma (HCC). However, several institutions have reported more comprehensive indication criteria for LT, with comparable survival outcomes. This literature review summarizes the recent updated on the indications for LT in patients with HCC. Several criteria expand the Milan criteria, which can be subdivided into the “based on tumor number and size only,” “based on tumor number and size plus tumor markers,” and “based on tumor differentiation” groups, with the outcomes comparable to those of the patients included within the Milan criteria. In addition to the tumor size and number, which are included in the Milan criteria; recent criteria include biomarkers and tumor differentiation. Several retrospective studies have reported microvascular invasion (MVI) as a significant risk factor for postoperative recurrence, highlighting the importance of preoperatively predicting MVI. Several studies attempted to identify preoperative predictive factors for MVI using tumor markers or preoperative imaging findings. Patients with HCC who are LT candidates are often treated while on the waiting list to prevent the progression of HCC or to reduce the measurable disease burden of HCC. The expanding repertoire of chemotherapeutic regiments suitable for patients with HCC should be further investigated.
- Table 1 is visually cluttered and somewhat difficult to read. Specifically, the "Criteria" column lacks clear differentiation from other studies. You could group similar criteria together for better visual organization.
Response: We thank the Reviewer for their comment. We have subdivided the criteria into the “based on tumor number and size only,” “based on tumor number and size plus tumor markers,” and “based on tumor differentiation” groups.
- The order of the eligibility criteria in Table 1 is unclear. Are the criteria arranged chronologically, alphabetically, or with/without biomarkers? Please specify the rationale behind this arrangement in the table legend. If no specific order is followed, reorganizing the criteria based on clinical relevance or frequency of use could enhance readability and utility.
Response: We thank the Reviewer for their comment. We have organized the table based on the year when the criteria were reported.
- Listing the first author in Table 1 adds limited value. Instead, prioritize summarizing clinically relevant data, such as the number of patients analyzed, outcomes, or the breakdown of DDLT vs. LDLT. If space is limited, it is advisable to omit the first author entirely to focus on more critical information.
Response: We thank the Reviewer for their comment. The first authors were removed from the table.
- The manuscript does not adequately address the prevailing trends in the use of eligibility criteria. To strengthen the discussion, provide an overview of the current global practices. Discuss which criteria are predominantly used today, with specific attention to regional variations—e.g., in Europe, North America, and Asia. Incorporating a clear summary of these trends, supported by relevant data or literature, would add substantial value to the manuscript.
Response: We thank the Reviewer for their comment. We have added the geographic distribution of the LT criteria as follows (page 6, lines 13–19): “The current European Association for the Study of the Liver guidelines and the American Association for the Study of Liver Diseases (AASLD) guidelines recommend the Milan criteria for the indication of LT [56, 57]. In the AASLD criteria, the UCSF criteria, total tumor volume cutoff of 115 cm, up-to seven criteria, extended Toronto criteria, and Kyoto criteria are also proposed as the expanded criteria [57]. In the guidelines for the Asia-Pacific region, the Milan criteria are also proposed and the expanded criteria are considered limited due to the scarcity of the donor pool [58].”